# Improving Interpretation Consistency of Serum Capillary Electrophoresis by Development of Quantitative Graphic Indexes

**DOI:** 10.3390/ijms252212240

**Published:** 2024-11-14

**Authors:** Jia-Ruei Yu, Yu-Tan Wu, Yi-Juan Sung, Tzong-Shi Chiueh, Wei-Hsuan Yu, Feng-Nan Hwang, Zong-Qi Wu, Zayd Anwar, Wan-Ying Lin, Hsin-Yao Wang

**Affiliations:** 1Department of Laboratory Medicine, Chang Gung Memorial Hospital, Linkou Main Branch, Taoyuan City 333, Taiwan; st33351995@gmail.com (J.-R.Y.); drche0523@gmail.com (T.-S.C.); 2Department of Mathematics, National Central University, Taoyuan City 320, Taiwan; wuyutan1125@gmail.com (Y.-T.W.); u_p54@ymail.com (Y.-J.S.); whyu@math.ncu.edu.tw (W.-H.Y.); hwangf@math.ncu.edu.tw (F.-N.H.); scwu1168@gmail.com (Z.-Q.W.); 3Department of Internal Medicine, Vassar Brothers Medical Center, Poughkeepsie, NY 12601, USA; zayd.anwar1@nuvancehealth.org; 4Department of Medicine, University of California San Diego, San Diego, CA 92093, USA; WAL018@health.ucsd.edu; 5School of Medicine, National Tsing Hua University, Hsinchu City 300, Taiwan

**Keywords:** capillary zone electrophoresis, monoclonal gammopathy, reference interval

## Abstract

Capillary zone electrophoresis–immunosubtraction (CZE-IS) is an essential laboratory test in diagnosing plasma cell neoplasms. However, the current interpretation of the test results is subjective. To evaluate CZE-IS in a more precise manner, this study proposed five key indexes, namely sharpness index, light chain index, immunoglobulin G index, immunoglobulin A index, and immunoglobulin M index. The reference intervals of these indexes were established using CZE-IS curve data from a clinical laboratory of a referral medical center. A total of 1000 cases with normal electrophoretic patterns were sampled for reference intervals establishment, and an additional 20 cases were included for validation. The following reference intervals in the γ zone were established: 1-6 (sharpness index), 1.06-2.71 (light chain index), 37-454 (immunoglobulin G index), (−9)-41 (immunoglobulin A index), and (−16)-46 (immunoglobulin M index). For the β2 zone, the reference intervals were 3-17 (sharpness index), 0.44-1.90 (light chain index), (−7)-61 (immunoglobulin G index), 2-117 (immunoglobulin A index), and (−12)-35 (immunoglobulin M index). The diagnostic performance of reference intervals of the proposed indexes in validation ranged from 95% to 100%. CZE-IS indexes provide the objective quantification of key characteristics of CZE-IS curves and improve the precision of CZE-IS interpretation.

## 1. Introduction

The identification of monoclonal gammopathy is essential in diagnosing various plasma cell neoplasms [1]. Capillary zone electrophoresis–immunosubtraction (CZE-IS) is a major laboratory method for identifying monoclonal gammopathy because of its established automation and short turnaround time [2,3]. However, the interpretation of CZE-IS results currently requires clinical pathologists’ manual review, which is subjective and lacks a common standard. Inconsistencies are common between different pathologists and even by the same pathologist over time. The CZE-IS is a qualitative test with a highly complicated and subjective interpretation. The inconsistency of CZE-IS interpretation primarily stems from the manual review of electrophoretic curves. The detection of monoclonal proteins is based on the superimposition of different curves. A curve with monoclonal gammopathy exhibits a sharp electrophoretic peak, which would be absent if the corresponding antisera is applied with the electrophoresis. Thus, different pathologists may have different standards regarding whether a peak is present or not. Furthermore, it is challenging for a pathologist to maintain a consistent interpretation standard. The shift of interpretation standards could lead to paradoxical serial results that do not match clinical manifestations [4]. In the worst-case scenario, this inconsistency would lead to incorrect clinical management and potential harm to patients.

Setting reference intervals for CZE-IS would be a solution to a more precise and objective interpretation of CZE-IS. Reporting reference intervals is a crucial method for aiding clinical interpretation. By comparing a patient’s laboratory test results with a given reference interval, clinicians gain better insight into the laboratory results. Moreover, reference intervals serve as a standard, improving interpretation consistency among clinicians. Typically, reference intervals are established to evaluate the concentration of an analyte in clinical laboratories. This concept can also extend to other clinical domains. A classic example is the electrocardiogram (ECG), where specific segments and intervals are defined, and their respective reference intervals are well-established [5]. Cardiologists comprehensively evaluate these values when interpreting an ECG and develop a corresponding differential diagnosis if there is an abnormal ECG index.

In this study, we aimed to develop key indexes to describe CZE-IS curves in a more objective way. The key indexes were defined in a CZE-IS curve, and their reference values were established. These CZE-IS indexes, derived from pathologists’ existing interpretation methods, include sharpness, light chain, immunoglobulin G (IgG), immunoglobulin A (IgA), and immunoglobulin M (IgM). These indexes represent the key information of an entire CZE-IS graph. By evaluating these indexes, pathologists can achieve a consensus interpretation of a normal CZE-IS pattern. In addition, clinicians can easily acquire a summary of a given CZE-IS curve. The interpretation of CZE-IS can be described and transmitted. The proposed indexes are believed to improve the quality of CZE-IS interpretation.

## 2. Results

### 2.1. Reference Interval Development and Validation

Figure 1, Figure 2 and Figure 3 present the illustration, reference intervals, and validation results of our five proposed indexes. All reference intervals underwent outlier detection, nonparametric reference interval establishment, and binomial validation testing. Only a minimal number of samples were removed during outlier detection, and all reference intervals were successfully validated. The median age of the included population is 65 years old, and the percentage of male and female cases were both 50%. Details regarding reference interval development and validation are summarized in Table A1.

Figure 1 illustrates the protein electrophoretic curves with different sharpness indexes. A higher sharpness index in a specific electrophoretic peak indicates a sharper peak. Thus, a high sharpness index in the γ or β2 zone suggests possible monoclonal gammopathy. The reference intervals for the sharpness index in the γ zone (Figure 1A) and the β2 zone (Figure 1B) are 1-6 and 3-17, respectively. The diagnostic performance of the validation was 100% for the γ zone and 95% for the β2 zone.

Figure 2 displays the protein electrophoretic curves with different light chain indexes. A light chain index outside the reference interval indicates an imbalance between κ and λ immunoglobulins, suggesting possible monoclonal gammopathy. The light chain index has a clinical meaning similar to serum-free immunoglobulin light chains. However, while serum-free immunoglobulin light chains mainly screen for light chain diseases, the light chain index screens for all types of monoclonal gammopathy. The reference intervals for the light chain index in the γ zone (Figure 2A) and the β2 zone (Figure 2B) are 1.06-2.71 and 0.44-1.90, respectively. The diagnostic performance of the validation was 100% for the γ zone and 95% for the β2 zone.

Figure 3 depicts protein electrophoretic curves with different immunoglobulin indexes. Immunoglobulin indexes include the IgG, IgM, and IgA indexes, which correlate with the amount of specific antibodies. A higher immunoglobulin index indicates hypergammaglobulinemia, which could be either monoclonal or polyclonal gammopathy. Conversely, a lower immunoglobulin index indicates hypogammaglobulinemia. Reference intervals for the IgG index were 37-454 in the γ zone (Figure 3A) and (−7)-61 in the β2 zone (Figure 3B). Reference intervals of the IgA index were (−9)-41 in the γ zone and 2-117 in the β2 zone. Reference intervals of the IgM index were (−16)-46 in the γ zone and (−12)-35 in the β2 zone. The diagnostic performance of the validation achieved 100% for reference intervals of the IgG and IgA index. For the IgM index, the performance was 95% for the γ zone and 100% for the β2 zone. Remarkably, the lower limits of some immunoglobulin indexes are negative. The immunoglobulin index is derived from the subtraction of two overlapping electrophoretic graphs, so negative immunoglobulin indexes would be possible if immunoglobulin levels are low. Negative immunoglobulin indexes are not associated with monoclonal proteins, and do not indicate further diagnostic examinations of plasma cell neoplasms.

Finally, Appendix A displays separate reference intervals of CZE-IS indexes by age and sex.

### 2.2. Case Demonstration

Figure 4 displays the CZE-IS graphs and CZE-IS indexes of three different cases. A case with normal immunoglobulin status would have all CZE-IS indexes within the reference intervals (Figure 4A). In contrast, a monoclonal gammopathy patient would exhibit an increased sharpness index in the γ or β2 electrophoretic zone. The light chain index either increases or decreases according to the light chain subtypes. Typically, the corresponding immunoglobulin index would be higher than the reference interval. However, this IgG-κ monoclonal gammopathy case has an IgG index within the reference range, indicating a milder monoclonal gammopathy (Figure 4B). Conversely, a patient with polyclonal gammopathy would have elevated immunoglobulin indexes, while the sharpness index and light chain index remain within their reference intervals (Figure 4C).

## 3. Discussion

This study defined the key indexes of CZE-IS results. Moreover, we established and validated the reference intervals of the indexes. With the indexes, the CZE-IS results can be described quantitatively. Interpretation of the CZE-IS would be more precise and communicable between pathologists. Thus, the proposed reference intervals aid in correctly identifying patients with monoclonal gammopathy.

The CZE-IS indexes correlate well with the relevant biochemical features and are explainable. In this study, the sharpness index and light chain index correlate with the clonality of immunoglobulins, and the immunoglobulin indexes correlate with the immunoglobulin concentrations. Monoclonal proteins are composed of immunoglobulins with identical protein structures and the same electrophoretic mobility. On the CZE-IS graph, the features of monoclonal proteins result in a single tall and sharp peak whose sharpness indexes are high. On the other hand, the light chain index reflects the proportion between κ and λ light chains in immunoglobulins and is similar to the κ/λ ratio in hematopathology [6]. An abnormal light chain index (both higher or lower) indicates abnormally increased κ or λ light chains. Finally, the immunoglobulin indexes correlate with the immunoglobulin concentrations. In protein electrophoresis, protein concentration can be quantified by the area under the electrophoretic peaks [7].

The CZE-IS indexes and their reference intervals reduce the imprecision of interpreting CZE-IS results. Typically, reporting a CZE-IS test differs significantly from traditional clinical biochemistry tests because a CZE-IS test requires clinical pathologists’ interpretation after the analytical process. This additional interpreting step introduces potential imprecisions. First, the definition of a monoclonal protein is not well-defined and dependent heavily on the interpreting pathologist [8]. Different clinical pathologists may reach different conclusions when interpreting the same CZE-IS result. Moreover, it is challenging for a clinical pathologist to maintain the same interpreting standard over time. The interpreting standard may vary slightly from day to day or even from sample to sample, causing within-run and between-run imprecision similar to traditional biochemistry tests. CZE-IS indexes and their reference intervals address this issue. For ambiguous CZE-IS curves, clinical pathologists can evaluate the peak by comparing the CZE-IS indexes with the well-defined reference intervals (Figure 5). This rigorous approach enhances the precision of CZE-IS interpretation. In addition, for serial CZE-IS follow-up tests, pathologists can better understand the disease course by monitoring changes in CZE-IS indexes. In terms of quality assurance, the CZE-IS indexes not only reduce imprecision but also aid in report verification. The CZE-IS indexes should be consistent with the final report. If discrepancies arise, pathologists should consider the possibility of erroneous reports and thoroughly rule out potential laboratory errors before finalizing the results.

The proposed CZE-IS indexes not only improve test precision but also aid in clinical interpretation. The CZE-IS indexes summarize the CZE-IS graph within a series of values. As Figure 4 displays, these index patterns reflect different kinds of globulin abnormalities. Clinicians can understand the information from the CZE-IS without interpreting the original graph by reading these indexes. This information is particularly helpful for junior clinicians or those unfamiliar with CZE-IS. In this study, five primary CZE-IS indexes were proposed, and they cannot cover all types of CZE-IS abnormalities. However, pathologists can develop similar indexes following the methods used in this study, allowing more various CZE-IS patterns to be evaluated simply. Notably, this study demonstrates the feasibility of establishing reference intervals for specific characteristics in laboratory graphs. Current guidelines typically discuss reference intervals for analyte concentrations, with no relevant discussion for laboratory graph characteristics. The guideline was extended to CZE-IS indexes, establishing reference intervals through a rigorous process followed by validation. This method of establishing reference intervals can be applied to other similar fields of laboratory medicine, such as other electrophoretic graphs, chromatographs, or mass spectrometry. This approach would significantly improve the quality of interpretation.

The CZE-IS indexes are different from machine learning (ML)-based interpreting methods [9,10,11]. Compared with ML methods, the reference interval-based method has several advantages. The explainability of the CZE-IS indexes is superior. These indexes are derived from the existing interpretation process, and the abnormal index patterns can correspond to different types of diseases. In addition, the concept of reference intervals is simple and straightforward for doctors. Clinicians are accustomed to reading laboratory data with reference intervals, and our proposed indexes align with this clinical practice. On the other hand, ML interpretation would directly provide a prediction of the CZE-IS curve. This prediction would have limited explainability, making it challenging for pathologists to integrate their interpretation with the ML prediction. Furthermore, establishing a reference interval requires fewer cases than ML-based methods, which is a practical consideration because the number of CZE-IS results is limited in most clinical laboratories. In this study, 1000 cases were used to establish the reference interval. However, according to CLSI EP28-A3c, establishing the reference interval only requires 120 cases, and validating existing proposed reference intervals only requires 20 cases. Conversely, training a good ML model would require significantly more cases, and validating an existing ML model lacks a consensual minimum case number. Compared to ML models, CZE-IS indexes are easier to transfer to other laboratories.

This study has some limitations. The indirect sampling method [12,13] was adopted to set up the reference intervals for the indexes. Patients’ medical conditions were not fully considered with the indirect sampling method. To establish reference intervals with a purely healthy population, the direct sampling method is worth using in future works. Additionally, like other biochemistry tests, the reference intervals proposed in this study are dependent on the population, electrophoretic instruments, and many other local factors. Therefore, to formally apply these results to other laboratories, a rigorous transference process is necessary. Our validation method described in the Methods section is one appropriate transference method, with more validating details available in CLSI EP28-A3c. Another limitation of the study is the lack of experimental controls to evaluate the efficacy of the proposed electrophoresis indexes. For the current stage of the proof-of-concept study, the proposed RI indexes were validated by using the binomial method, the recommended validation method endorsed by CLSI EP28-A3c. While the approach proposed in the study is the first-in-class quantitative method for electrophoresis test, further comparison study between the current qualitative approach (physicians’ interpretation without the RI indexes) and the proposed approach is worthy of investigation in the future.

In conclusion, this study defined the sharpness index, light chain index, and IgG/IgA/IgM index of CZE-IS and established their reference intervals. The reference intervals of these indexes provide an objective and describable standard for CZE-IS. The proposed CZE-IS indexes would improve the consistency of interpretation of CZE-IS.

## 4. Materials and Methods

### 4.1. Study Design and Environmental Settings

Figure 6 illustrates the process flowchart of this study. Several key measurements were defined based on the existing interpreting experience. Subsequently, the reference intervals of these indexes were established and validated according to current guidelines. Finally, how these indexes aid in CZE-IS interpretation was demonstrated using real-world case applications.

All experiments were conducted on a personal computer running Windows 10, with 24 GB of random-access memory and a 2.5 GHz Intel Core i5-12500H central processing unit (Intel, Santa Clara, CA, USA). The data were read and converted using Python 3.11.4, with the packages “pypyodbc”, “os”, “subprocess”, “csv”, “pandas”, “functools”, and “openyxl”. The index calculations were implemented using R 4.3.1, with the external packages “ggplot2”, “Matrix”, and the library “pracma”.

### 4.2. Capillary Zone Electrophoresis–Immunosubtraction Data

All data for the original CZE-IS curves in this study were obtained from the clinical laboratory in the Department of Laboratory Medicine at Chang Gung Memorial Hospital Linkou Main Branch, Taoyuan City, Taiwan, from April to September 2023. The CZE-IS test was performed using the Sebia Capillarys 3 Tera automated instrument (Sebia, Lisses, France). All data were directly derived from the analytical instrument, with no human specimens or clinical data collected. The study was conducted in accordance with the Declaration of Helsinki, and approved by the Institutional Review Board of Chang Gung Memorial Hospital (202401438B0, approval date: 18 September 2024).

The CZE-IS graph is a plot with molecular migration time on the *x*-axis. The absorbance measured by the optical detect system is plotted on the *y*-axis, typically representing protein concentration. In Sebia Capillarys 3 Tera, each piece of original CZE-IS data was represented as a 6 × 300 matrix. The six rows of numbers corresponded to six different CZE-IS curves run with distinct reagents (namely anti-IgG, anti-IgA, anti-IgM, anti-κ, and anti-λ antisera, and the original electrophoresis solution without antisera). The 300 data points represented the ultraviolet absorbance at 200 nm within a 300 s interval.

To establish the reference intervals, 1000 CZE-IS cases from April to August 2023 were randomly sampled, and an additional 20 cases in September 2023 were randomly sampled as reference individuals for validation. In this study, the reference intervals were established using the indirect sampling technique [14]. The inclusion of CZE-IS data was based on the curve pattern rather than patient characteristics. Curves without γ-globulin abnormalities were included, while curves with monoclonal gammopathy, polyclonal gammopathy, or other overt abnormalities were excluded. All aforementioned reference cases were interpreted by at least two different clinical pathologists.

### 4.3. Characteristic Indexes

#### 4.3.1. Sharpness Index

Monoclonal and polyclonal proteins have different peak patterns in the CZE-IS. The monoclonal proteins are composed of immunoglobulins with identical protein structures, so they would have the same electrophoretic mobility in CZE-IS. This results in a tall, sharp peak in the CZE-IS graph, while a polyclonal mixture of serum immunoglobulins would present a broad-based peak.

To address this, the “sharpness index” was proposed for a given electrophoretic peak. This index is derived from the second derivative of a function, which visually correlates with the peak sharpness of a curve. Given that CZE-IS data is composed of discrete, serial data points, the central finite difference method [15] was applied to calculate the sharpness index. The sharpness index of a CZE-IS curve at a given time x is defined as follows:f″(x) ≈ − [P(x + 1) + P(x − 1) − 2P(x)](1)
where P(x) represents the electrophoresis run without antisera. Here, the sharpness index was defined as the negative value of the second derivative. By this definition, the sharpness index of the peaks would present with a positive value, and a sharper peak correlates to a higher index.

To distinguish between monoclonal and polyclonal gammopathies, we focused only on the sharpness index of the peaks in the β2 and γ zones in this study. Figure 1 illustrates the sharpness index in the γ zone (Figure 1A) and the β2 zone (Figure 1B).

#### 4.3.2. Light Chain Index

In addition to the sharpness index, the clonality of proteins can also be distinguished by the distribution of light chain classes. In this study, the “light chain index” was proposed to evaluate the clonality of immunoglobulin.

For a given time x in the CZE-IS curve, the light chain index Q(x) is defined as follows:Q(x) = (P(x) − K(x))/(P(x) − L(x))(2)
where P(x), K(x) and L(x) represent the absorbance at time x in the electrophoretic procedure run with original electrophoresis solution, anti-κ, and anti-λ antisera, respectively.

This study focused on the light chain index of electrophoretic peaks in both the β2 zone and the γ zone. The light chain index of an electrophoretic peak reflects the ratio between immunoglobulins with κ light chain and λ light chain. Thus, an imbalanced light chain ratio indicates monoclonality. Figure 2 displays the illustration of the light chain index in (A) the γ zone and (B) the β2 zone.

#### 4.3.3. IgG/IgA/IgM Index

The increase in immunoglobulins is a significant indicator for diagnosing plasma cell neoplasms. While CZE-IS does not formally report quantitative information, a stronger electrophoretic signal correlates with a higher protein concentration according to the principles of electrophoresis.

The IgG, IgA, and IgM indexes (denoted as ΔG(x), ΔA(x) and ΔM(x), respectively) at a given x-coordinate x are defined by the following equations:ΔG(x) = P(x) − G(x)(3)
ΔA(x) = P(x) − A(x)(4)
ΔM(x) = P(x) − M(x)(5)
where P(x), G(x), A(x) and M(x) denote the absorbance at time x in the electrophoretic procedure run with original electrophoresis solution, anti-IgG, anti-IgA, and anti-IgM antisera, respectively.

This study focused on the immunoglobulin indexes of electrophoretic peaks in both the β2 zone and the γ zone. A higher immunoglobulin index indicates a higher concentration of the corresponding immunoglobulin. Figure 3 depicts the derivation of the IgG index in (A) the γ zone and (B) the β2 zone.

### 4.4. Reference Interval

#### 4.4.1. Development of the Reference Intervals for the Indexes

This study established a total of ten reference intervals. The process of reference interval development includes case inclusion, outlier detection, reference limit determination, and confidence interval establishment. The entire process adheres to the Clinical and Laboratory Standards Institute (CLSI) guideline EP28-A3c [14]. This section focuses on outlier detection and reference interval establishment. Details regarding case inclusion are available as previously mentioned in Section 4.2.

For outlier detection, the “one-third rule for the ratio D/R” [16,17], a common method supported by the CLSI working group, was applied. In this method, D is defined as the absolute difference between an extreme observation (either large or small) and its adjacent observation, and R is defined as the range of the samples. This method excludes a sample if the D/R ratio of the extreme sample exceeds one-third. In cases of outlier elimination, the sample number was complemented back to 1000.

To establish the reference intervals of the indexes, the 2.5th and 97.5th percentiles of the reference intervals were acquired using the nonparametric method [14]. In addition to the reference intervals, this study also established the 90%, two-tailed confidence intervals of both limits of the reference interval, using the same nonparametric method.

#### 4.4.2. Validation of the Reference Intervals for the Indexes

The proposed reference intervals were validated using the binomial method, the most well-documented method for validating a reference interval according to CLSI EP28-A3c [14]. The binomial method validates the reference interval by examining an additional small number of reference individuals. The established reference interval is considered valid if no more than 10% of the reference individuals fall outside the established limits. The reference interval is considered invalid if above 20% of the reference individuals fall outside the established limits. The validation result between both aforementioned conditions indicates a further testing process. In this study, a total of 20 cases were randomly sampled as reference individuals for validation. Appendix A illustrates the complete validation flow diagram.

## 5. Conclusions

This study defined the sharpness index, light chain index, and IgG/IgA/IgM index of CZE-IS and established their reference intervals. The diagnostic performance of the graphic indexes in validation was 95–100%. These indexes present different patterns in various diseases. The reference intervals of these indexes provide an objective and describable standard for CZE-IS to improve the quality and consistency of CZE-IS interpretations.

## Figures and Tables

**Figure 1 ijms-25-12240-f001:**
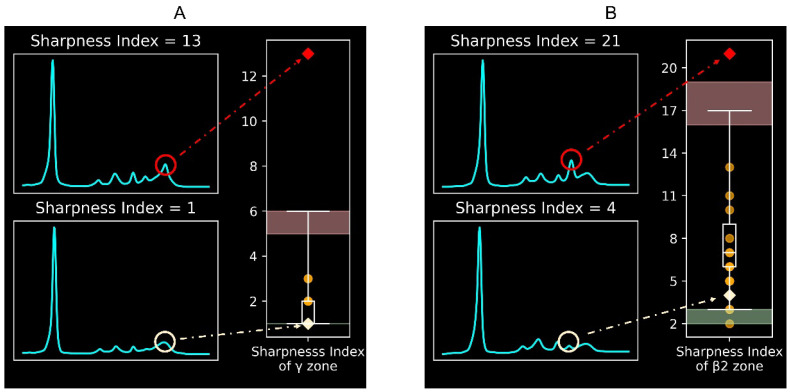
Illustration and reference intervals of the sharpness index in (**A**) the γ zone and (**B**) the β2 zone. A sharpness index higher than the reference interval indicates a sharper electrophoretic peak and implies monoclonal gammopathy. In the boxplots, the lower and upper black bars indicate the normal lower and upper limits of the reference interval. Red and green backgrounds indicate the confidence intervals of the upper and lower limits, respectively. Orange data points refer to the 20 cases used for validation. Red and beige data points refer to the two demonstration cases.

**Figure 2 ijms-25-12240-f002:**
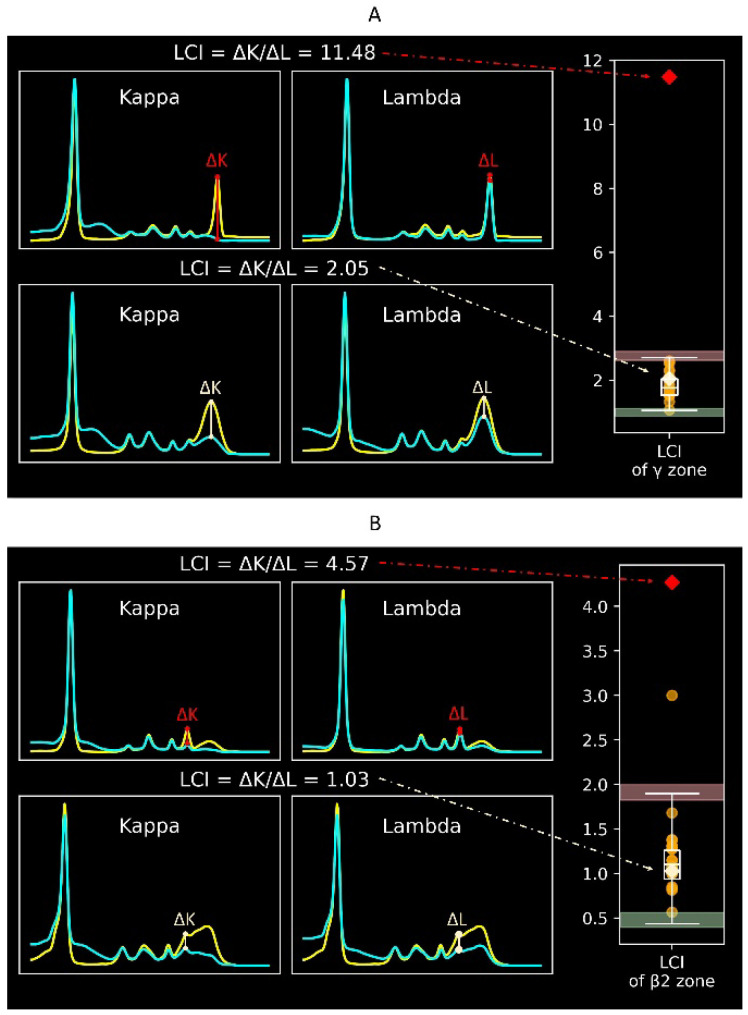
Illustration and reference intervals of the light chain index in (**A**) the γ zone and (**B**) the β2 zone. A light chain index falling outside the reference interval indicates light chain restriction and implies monoclonal gammopathy. In the subplots with two electrophoretic curves, the yellow curve indicates the original electrophoresis, and the blue curve indicates the electrophoresis run with antisera. In the boxplots, the lower and upper black bars indicate the normal lower and upper limits of the reference interval. Red and green backgrounds indicate confidence intervals of the upper and lower limits, respectively. Orange data points refer to the 20 cases used for validation. Red and beige data points refer to the two demonstrated cases.

**Figure 3 ijms-25-12240-f003:**
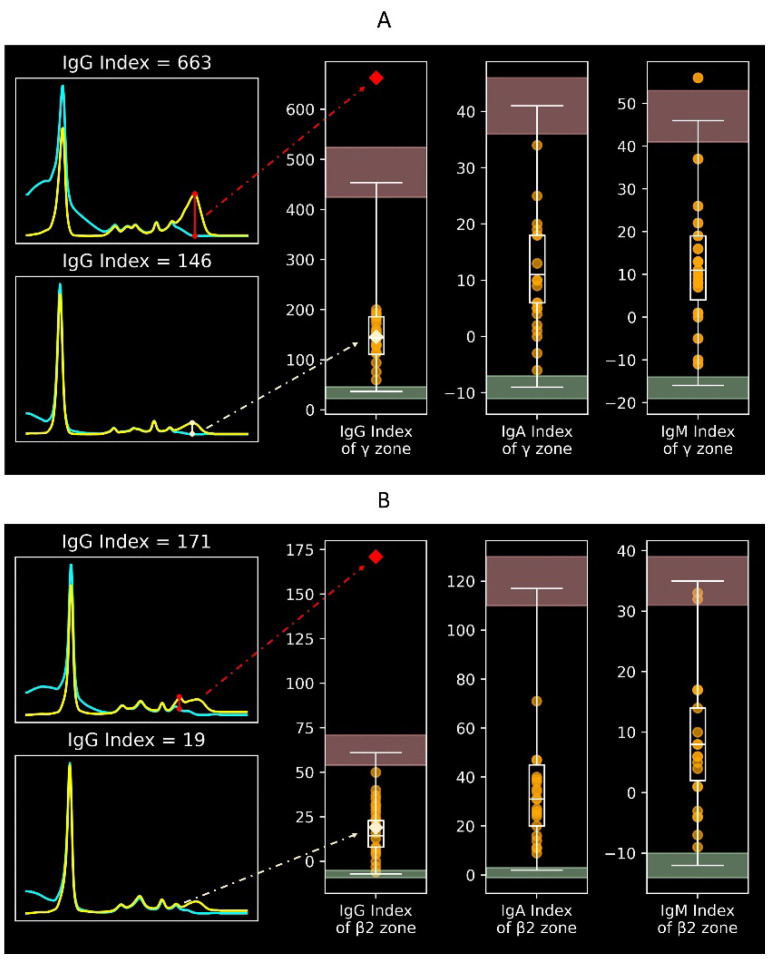
Illustration and reference intervals of the immunoglobulin indexes in (**A**) the γ zone and (**B**) the β2 zone. An immunoglobin index higher than the reference interval indicates an increase in immunoglobulin concentration. In the subplots with two electrophoretic curves, the yellow curve indicates the original electrophoresis, and the blue curve indicates the electrophoresis run with antisera. In the boxplots, the lower and upper black bars indicate the normal lower and upper limits of the reference interval. Red and green backgrounds indicate the confidence intervals of the upper and lower limits, respectively. Orange data points refer to the 20 cases used for validation. Red and beige data points refer to the two demonstrated cases.

**Figure 4 ijms-25-12240-f004:**
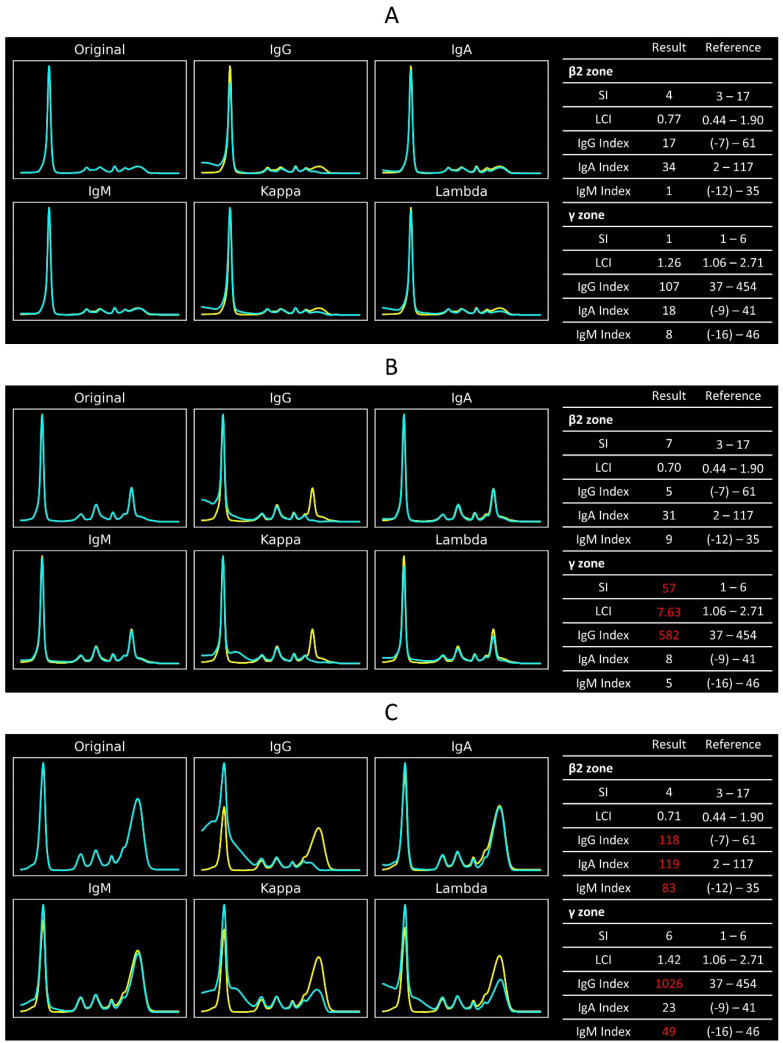
CZE-IS graphs and their respective CZE-IS indexes of (**A**) a normal case, (**B**) IgG-κ monoclonal gammopathy, and (**C**) IgG polyclonal gammopathy. In the subplots with two electrophoretic curves, the yellow curve indicates the original electrophoresis, and the blue curve indicates the electrophoresis run with antisera. The red CZE-IS index numbers indicate the results outside their reference intervals.

**Figure 5 ijms-25-12240-f005:**
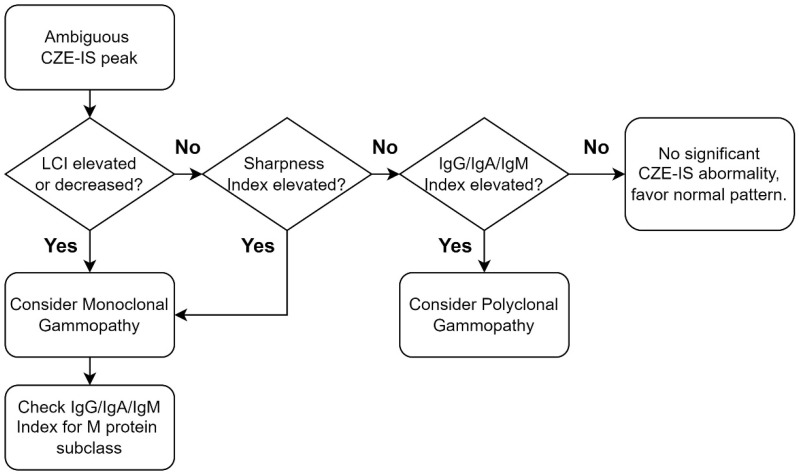
Proposed diagnostic algorithm for ambiguous CZE-IS peaks using CZE-IS indexes. Abbreviations: CZE-IS, capillary zone electrophoresis–immunosubtraction; LCI, light chain index; IgG, immunoglobulin G; IgA, immunoglobulin A; IgM, immunoglobulin M.

**Figure 6 ijms-25-12240-f006:**
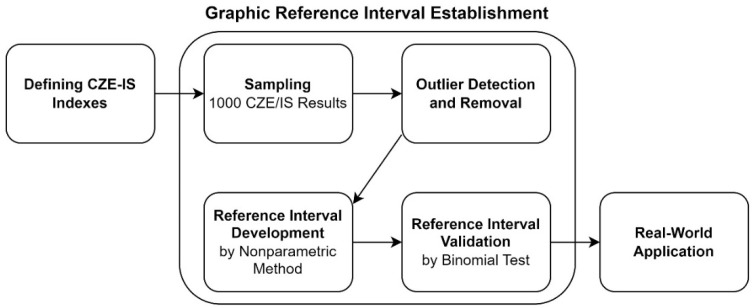
Process flowchart of this study.

## Data Availability

The data presented in this study are available on request from the corresponding author. The data are not publicly available due to privacy.

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
