# Peer review of "Improving Interpretation Consistency of Serum Capillary Electrophoresis by Development of Quantitative Graphic Indexes"

_ijms, 2024, doi:10.3390/ijms252212240_

Round 1
Reviewer 1 Report
Comments and Suggestions for Authors
- Overall, the paper is well written and present quite orignal data to improve the diagnosis of gammapathies.
- My main concern is related to the way RI were determined: How can you be sure that patients who were used for RI were normals? (i.e. without gammapathy?). The risk is to include pathological patients who could increase the higher limit of RI for exemple. Exclusion of patients with anemia? Renal failure? History of myeloma? Healthy volunteers?... Not clear enough for RI determination. There is also no data about the age of the population and sex repartition. Please add these data.
- Minor comment: Why results are presented before M&M?
Reviewer 2 Report
Comments and Suggestions for Authors
This paper presents the development and validation of new quantitative graphic indexes (CZE-IS indexes) aimed at improving the consistency of interpretation in capillary zone electrophoresis-immunosubtraction (CZE-IS). The authors propose five major indexes—sharpness index, light chain index, immunoglobulin G (IgG) index, immunoglobulin A (IgA) index, and immunoglobulin M (IgM) index—and use them to evaluate CZE-IS results. The reference intervals for these indexes were established using 1000 normal electrophoretic patterns, and the results of validation with 20 cases show that these reference intervals were accurately established, demonstrating high diagnostic precision.
I believe this study is meaningful as it provides a new standard to simplify and clarify the interpretation of CZE-IS results, which is particularly important for the diagnosis of monoclonal gammopathies, such as multiple myeloma. By reducing the variability in interpretation between pathologists and improving diagnostic accuracy in clinical practice, this study has significant value.
However, the following points need to be addressed:
Comments:
1.Lack of Explanation for Axes in CZE-IS Graphs
In the CZE-IS graphs, it is not clearly stated what the vertical and horizontal axes represent. If the vertical axis represents protein concentration or peak intensity, and the horizontal axis indicates molecular migration distance or time, it is important to explicitly describe these physical quantities. This clarification will help readers properly interpret the graphs.
2.Unclear Relationship Between CZE-IS Indexes and Physical Phenomena
While the effectiveness of the proposed CZE-IS indexes is demonstrated, it is not clearly explained how these indexes relate to specific substances or physical phenomena. For example, how the sharpness index corresponds to changes in protein concentration or migration speed, or how the immunoglobulin indexes relate to specific molecular behaviors, should be detailed based on experimental results and references. Providing this explanation would clarify the connection between physical quantities and the data, allowing readers to gain a deeper understanding.
By addressing these points, the scientific value of this research can be further enhanced, and the practical utility of CZE-IS interpretation will be improved.
